# Olive Oil Improves While Trans Fatty Acids Further Aggravate the Hypomethylation of LINE-1 Retrotransposon DNA in an Environmental Carcinogen Model

**DOI:** 10.3390/nu14040908

**Published:** 2022-02-21

**Authors:** Laszlo Szabo, Richard Molnar, Andras Tomesz, Arpad Deutsch, Richard Darago, Timea Varjas, Zsombor Ritter, Jozsef L. Szentpeteri, Kitti Andreidesz, Domokos Mathe, Imre Hegedüs, Attila Sik, Ferenc Budan, Istvan Kiss

**Affiliations:** 1Department of Public Health Medicine, Medical School, University of Pécs, 7624 Pécs, Hungary; laszlo.szabo.pte@gmail.com (L.S.); richard.molnar.pte@gmail.com (R.M.); andras.tomesz.pte@gmail.com (A.T.); deutsch.arpad@pte.hu (A.D.); richard.darago.pte@gmail.com (R.D.); varjas.timea@pte.hu (T.V.); budan.ferenc@pte.hu (F.B.); kiss.istvan@pte.hu (I.K.); 2Department of Medical Imaging, Medical School, University of Pécs, 7624 Pécs, Hungary; ritterzsombor@gmail.com; 3Institute of Transdisciplinary Discoveries, Medical School, University of Pécs, 7624 Pécs, Hungary; sik.attila@pte.hu; 4Department of Biochemistry and Medical Chemistry, University of Pécs Medical School, 7624 Pécs, Hungary; andreidesz.kitti@pte.hu; 5Department of Biophysics and Radiation Biology, Faculty of Medicine, Semmelweis University, 1085 Budapest, Hungary; hegedus.imre1@med.semmelweis-univ.hu; 6Hungarian Centre of Excellence for Molecular Medicine, In Vivo Imaging Advanced Core Facility, 1094 Budapest, Hungary; 7Institute of Physiology, Medical School, University of Pécs, 7624 Pécs, Hungary

**Keywords:** extra virgin olive oil, trans fatty acid, DMBA, LINE-1 methylation pattern

## Abstract

DNA methylation is an epigenetic mechanism that is crucial for mammalian development and genomic stability. Aberrant DNA methylation changes have been detected not only in malignant tumor tissues; the decrease of global DNA methylation levels is also characteristic for aging. The consumption of extra virgin olive oil (EVOO) as part of a balanced diet shows preventive effects against age-related diseases and cancer. On the other hand, consuming trans fatty acids (TFA) increases the risk of cardiovascular diseases as well as cancer. The aim of the study was to investigate the LINE-1 retrotransposon (L1-RTP) DNA methylation pattern in liver, kidney, and spleen of mice as a marker of genetic instability. For that, mice were fed with EVOO or TFA and were pretreated with environmental carcinogen 7,12-dimethylbenz[a]anthracene (DMBA)—a harmful substance known to cause L1-RTP DNA hypomethylation. Our results show that DMBA and its combination with TFA caused significant L1-RTP DNA hypomethylation compared to the control group via inhibition of DNA methyltransferase (DNMT) enzymes. EVOO had the opposite effect by significantly decreasing DMBA and DMBA + TFA-induced hypomethylation, thereby counteracting their effects.

## 1. Introduction

Adverse environmental effects often cause epigenetic modifications. In turn, the resulting genomic instability and abnormal methylation patterns can be observed in the background of cardiovascular and malignant diseases, obesity, type 2 diabetes mellitus, and neurodegenerative diseases [1]. A good example is the dietary intake of trans fatty acid (TFA), mainly from hydrogenated fats, which can account for 0.2–6.5% of energy intake [2]. In countries where more olive oil (OO) is consumed as an alternative to hydrogenated fats, the damage caused by TFA is lower [3].

Early epigenetic alterations may usually be reversed through chemopreventive compounds according to clinical trials [4]. Nutritional factors are the most important in chemoprevention and exert their effects mainly through antioxidation and anti-inflammatory effects. Anticancer effects of nutrition are mediated partly through gene expression ensured by genomic stability. Proapoptotic effects or antiproliferative regulation by nutritional factors can help to maintain genomic stability, as supported by numerous in vitro experiments, in vivo experiments, and clinical trials [5,6,7,8,9,10]. The frequent consumption of OO, particularly extra virgin olive oil (EVOO), has been shown to protect against cardiovascular diseases and malignancies and even to increase life expectancy [11,12]. The constituents of OO are capable to reduce the infarct size, exert strong antioxidant protection, and reduce the total cholesterol as well as triglyceride level in vivo [13]. The aging process is also accompanied by epigenetic and gene expression changes, mainly due to alterations in DNA methylation patterns toward a genome-wide more hypomethylated state [14].

### 1.1. Effects of Trans-Fatty Acids

TFA content of food increases the risk of cardiovascular diseases (CVD), breast cancer, prostate cancer, diabetes, and obesity [15], which also shortens life expectancy. A 16-year prospective cohort study in the United States analyzed the fat intake of 521,120 people [16]. The limits of the quintiles of amounts of daily TFA intakes were 1.41; 1.81; 2.2; and 2.73 percentages of calorie intake. Between the data from the upper and lower quintiles is a positive association with mortality based on gender- and age-normalized hazard ratio (1.03; CI 1.00–1.05; *p* trend = 0.0062) [16].

In a meta-analysis, 7 prospective studies of total dietary TFA intake and 5 studies of serum TFA in which participants were 26 years old or older and appeared to be healthy were analyzed. Although TFA intake does not correlate with overall cancer mortality, a positive association between dietary TFA intake and relative risk (RR) of breast cancer (1.37; 95% CI 1.04–1.81; *p* = 0.02) was found in postmenopausal women [17]. Another meta-analysis involving nearly 140,000 subjects demonstrated the adverse effect of TFA, namely a 2% energy intake increase in dietary intake of TFA significantly elevated the risk of cardiovascular disease (RR 1.23 95% CI 1.11–1.37; *p* < 0.001) [18].

Thus, TFA-induced damages increase the risk of cardiovascular diseases (CVD), breast cancer, prostate cancer, diabetes, and obesity [15], through which TFA presumably also shortens life expectancy. In contrast, Alfin-Slater and coworkers fed rats with a TFA-enriched diet (TFA content was 0.32% of the body weight, 30 times of the human consumption/kg of body weight) and found no difference in life expectancy between rats fed with this diet and rats fed with a conventional diet [19].

However, TFA damage may be especially harmful through enhancing transforming growth factor-beta (TGF-β) production in case of solitary fibrous tumors, neoplasms (angiomyolipoma, leiomyoma, hemangioma, lymphangioma, juxtaglomerular cell tumor, renomedullary interstitial cell tumor, lipoma, and schwannoma), and malignant tumors (leiomyosarcoma, rhabdomyosarcoma, angiosarcoma, osteosarcoma, synovial sarcoma, fibrosarcoma, malignant fibrous histiocytoma) arising from renal mesangial cells [20].

### 1.2. Effects of Olive Oil

EVOO has 55–83% of omega-9 oleic acid, which is a monounsaturated fatty acid (MUFA), 3.5–21% of linoleic acid, which is a polyunsaturated fatty acid (PUFA), 7.5–20% of saturated palmitic acid, and 0.5–5% of stearic acid content, while triunsaturated omega-3 α-linolenic acid is present in 0–1.5% [21]. In addition, EVOO also contains protective water-soluble substances, the best-known being oleuropein and oleocanthal [22,23].

A meta-analysis by Pelucchi and coworkers found based on five case-control studies that the pooled RR of breast cancer between the lowest and the highest quartiles of the population consuming a diet containing olive oil was 0.62 (95% CI 0.44–0.88) [24]. In another study, it was found that OO consumption significantly reduced the risk of the development of lung cancer (OR: 0.65; 95% CI: 0.42–0.99; *p* < 0.05) [25]. A case-control study found a significant difference in the protective effect against laryngeal cancer between the highest quartile consuming 42.9 g olive oil per day and the lowest quartile consuming less than 3.2 g per day (OR: 0.4 (95% CI: 0.3–0.7; *p* = 0.01) [26]. In a case-control study, a statistically significant inverse dose–response relationship was also found between the risk of developing bladder cancer and the level of olive oil consumption, when comparing the data of the lower tertile and middle tertile of less than 1.6 g OO consumption per day (OR: 0.62; 95% CI: 0.39–0.99) and the data of the lower and upper tertile consuming over 3.9 g per day (OR: 0.47; 95% CI: 0.28–0.78; *p*-trend = 0.002) [27].

### 1.3. The Effect of DMBA

The environmental carcinogen 7,12-dimethylbenz[a]anthracene (DMBA) is a harmful substance that can also be found in exhaust fumes, tobacco smoke, and burnt food. DMBA increased the risk of the development of bladder cancer, skin cancer, and soft tissue malignancies in proportion to age in rodents [28]. Thus, the DMBA-induced changes in molecular epidemiological biomarkers can reliably predict both the adverse environmental effects and the protective effect of chemopreventive agents, on which animal models can be based [29,30].

According to our present knowledge, no data are available either on the annual global exposure of humans to DMBA or on its effects on reducing life expectancy, but DMBA damage causes LINE-1 retrotransposon (L1-RTP) DNA hypomethylation [31], which is a relevant biomarker of biological aging [14,32].

### 1.4. DNA Methylation

DNA methylation involves the substitution of the hydrogen atom of the number 5 carbon atom of the cytosine ring by a methyl group due to the action of DNA methyltransferase (DNMT) enzymes. This epigenetic regulatory mechanism silences the gene expression of the given gene by methylation at the tandem repeating CpG (cytosine preceding guanosine) islands in the promoter and/or enhancer region of genes [14,31]. The L1-RTP DNA methylation pattern is a representative biomarker of global methylation, with positive correlations between them [31,33].

Hypomethylation may be induced/caused by passive demethylation of DNA, lack of methyl-donor group containing substrates (for example methionine-deficient diet) [34], or by the altered functioning of DNMT enzymes [35]. The activity of DNMT enzymes is generally reduced in global hypomethylation, but parallelly, the activity of the DNMT1 enzyme may increase, leading to the hypermethylation of CpG islands of the tumor suppressor genes, silencing them—and thus increases the risk of carcinogenesis or malignancy [36].

Different organs in vivo and tumor cells in vitro show various correlation patterns between their aging and the possibility of the occurrence of mutations in them [37,38,39,40]. Mahmood and coworkers have found a positive association between L1-RTP hypomethylation measured in the DNA content of the cell-free fraction of blood and aging and the increased likelihood of malignant tumorous diseases [41,42]. Interestingly, however, the correlation between aging and the probability of development of somatic mutations in the kidney renal cell carcinoma (KIRC) and the kidney renal papillary cell carcinoma (KIRP) cell lines is inverse (Horvath, 2013)—which provides a basis for studying the methylation pattern of renal DNA.

#### 1.4.1. DNA Methylation and Malignant Tumors

Global DNA hypomethylation occurs in malignant tumor tissues, but this is not a permanent process but a sudden one, usually preceding malignant transformation (Sheaffer, 2016). For example, there is a significant (*p* < 0.001) correlation between the incidence of hepatocellular carcinoma and the hypomethylation of serum L1-RTP DNA [43]. According to published results *C-MYC* gene expression—which is also important in carcinogenesis—increased with aging due to the hypomethylation of the promoter region in both the spleen and the liver of mice—and this may also cause downregulation of P53, which protects against aging through treating DNA damage [44,45,46].

#### 1.4.2. DNA Methylation and Aging

With aging global DNA methylation levels tend to decline continuously—this phenomenon is known as “epigenetic drift” [14,28,35], which is also strongly influenced by environmental factors [38,47]. On the other hand, the “epigenetic clock” represents with respect to specific DNA segments and organs, how methylation of CpG regions changes with aging [Horvath, 2013; Jones, 2015; Lim, 2018]. For example, aging correlates with the hypomethylation of the liver tissue DNA both in human and rodent liver [48].

Obesity, smoking, and the lack of exercise are also positively associated with L1-RTP DNA hypomethylation in white blood cells [49] and with reduced life expectancy [47]. In Europe, smoking shortened life expectancy on average by 19.8% in men and by 18.9% in women, and overweight and obesity by 7.7% in men and by 11.7% in women [50].

### 1.5. Objective

Our study aimed to examine the L1-RTP DNA methylation pattern in the liver, spleen, and kidneys of DMBA-treated TFA—and EVOO-fed mice to determine whether the change in the quantitative values compared to the DMBA-treated and the controls free of DMBA reflects the expected harmful effect of TFA and the protective effect of EVOO, as reported in the literature. Furthermore, we also examined whether the change in the L1-RTP DNA methylation pattern was associated with the predictors of life expectancy of dietary TFA and EVOO consumption, and with DMBA exposure, based upon literature data.

A further aim of the experiment was to determine whether the effects of these carcinogenic/chemopreventive agents could be examined with the L1-RTP DNA methylation pattern as potentially relevant biomarker.

Our present study aims to investigate the extent of L1-RTP DNA methylation on the effects of DMBA exposure combined with TFA or EVOO consumption in the liver, spleen, and kidneys of mice in vivo.

## 2. Materials and Methods

We used eight groups of 12-week-old female CBA/Ca mice (*n* = 6) in our study. Untreated control and DMBA-treated control groups received no prefeeding, while one group of animals received 300 mg/day/animal of olive oil (Agraria Riva Del Garda SCA) and 300 mg/day/animal of TFA (trans-3-hexadecenoic acid) (Sigma Aldrich), respectively, in addition to their usual diet for 2 weeks before DMBA treatment. Table 1. contains the exposure details for DMBA, TFA and olive oil.

Apart from the untreated (negative control) control group, the other seven groups received 20 mg/kg bodyweight DMBA intraperitoneally (Sigma-Aldrich) dissolved in 0.1 mL of corn oil. The negative control group was also injected with 0.1 mL corn oil. (Although the corn oil contains chemopreventive linoleic acid in 58–62% in earlier experiments, the effect of DMBA was proper, or even due to *n*-6 essential fatty acid content it could enhance the effect of DMBA [51,52,53]. After 24 h of DMBA exposure, the organs to be tested (liver, kidneys, and spleen) were removed after cervical dislocation.

Mice were housed according to the principles and guidelines of animal experimentation. Every effort was made to minimize their suffering. The experiment was conducted by following the ethical standards in force (University of Pécs, Animal Welfare Committee; Ethical approval number: BA02/2000-79/2017).

### 2.1. Isolation of DNA

DNA was isolated using the High Pure PCR Template Preparation Kit (Roche, Madison, WI, USA) according to the manufacturer’s instructions.

### 2.2. LINE-1 DNA Methylation

We used EpiTect Bisulfite kit (Qiagen, Hilden, Germany) for bisulfite conversion according to the manufacturer’s instructions. This process resulted in the conversion of unmethylated cytosines into uracil. High-resolution melting (HRM) analysis was then performed, which, based on the difference in melting point, was able to distinguish between uracil and methylated cytosine bases. If the DNA contains highly methylated regions, bisulfite conversion and subsequent amplification will result in a higher melting point because the retention of more cytosine will result in a higher GC content of the amplified fragment (there are three hydrogen bonds between guanine and cytosine). In less methylated regions, unmethylated cytosines are converted to adenine resulting in a lower melting temperature.

For the HRM analysis, primers targeting the CpG-rich region of LINE-1 were used [Newman, 2012], and the sequences were as follows: forward: 5′-GGT TGA GGT AGT ATT TTG TGT G-3′, reverse: 5′- TCC AAA AAC TAT CAA ATT CTC TAA C-3′. Amplification was performed in 96-well plates in a Roche LightCycler480 qPCR instrument (Roche, Madison, WI, USA). The reaction mix contained 20 ng of bisulfite-treated DNA, 0.75-0.75 μM forward and reverse primers, 1xLightCycler 480 High Resolution Melting Master (Roche, Madison, WI, USA) in 20 μL final volume [Bray, 2018)]. The parameters of PCR were the following: heating to 95 °C for 5 min, followed by 35 cycles: 1. 95 °C for 20 s, 2. 60 °C for 30 s, 3. 72 °C for 20 s. Then melting point/melting curve analysis was performed between 73 °C and 84 °C with temperature steps of 0.1 °C/2 s.

We used mouse high methylated genomic DNA (EpigenDx, Hopkinton, MA, USA) and mouse low methylated genomic DNA (EpigenDx, Hopkinton, MA, USA) for positive and negative controls, respectively, and their mixtures in different proportions to allow quantification of the methylation level of our samples.

### 2.3. Calculation and Statistical Analysis

We calculated and compared the relative L1-RTP DNA methylation levels of L1-RTP DNA expression levels using the 2-ΔΔCT method. The Kolmogorov–Smirnov test was used to examine the distribution of the results and Levene’s *F*-test and *T*-test were used to compare means. Calculations and analyses were performed using IBM SPSS 21 statistical software and the level of statistical significance was set at a *p*-value of <0.05.

Average DNA methylation levels were expressed as the percentage of untreated animals (negative controls).

## 3. Results

Compared to untreated control, EVOO coadministered with DMBA could partly ameliorate the hypomethylating effect of DMBA. DMBA alone and DMBA + TFA-induced significant L1-RTP DNA hypomethylation in the spleen (Figure 1).

DMBA and DMBA + TFA-induced significant L1-RTP DNA hypomethylation in the liver, compared to the negative control. EVOO ameliorated the effect of DMBA (Figure 2).

EVOO coadministered with DMBA could partly ameliorate the hypomethylating effect of DMBA. DMBA alone and DMBA + TFA caused significant L1-RTP DNA hypomethylation in the kidneys (Figure 3). 

Thus, our observations showed that DMBA administered alone induced statistically significant L1-RTP DNA hypomethylation in all organs examined. DMBA could induce only a small, statistically not significant hypomethylation in all the three organs examined, if protective EVOO was added as well. As we expected, the combined effect induced by TFA and DMBA was significant and the highest degree of L1-RTP DNA hypomethylation in all three organs was observed. The numerical results of methylation level measurements are found in Appendix A’s Table A1, Table A2 and Table A3.

## 4. Discussion

Both DMBA and TFA generate ROS with partly overlapping molecular effects and signal transduction mechanisms [44,54,55].

### 4.1. Effect of ROS on the L1-RTP DNA Methylation and Aging

The damage caused by reactive oxygen species (ROS) generated during the decay of DMBA and TFA mainly contributes to global hypomethylation [44,54,55]. ROS depletes glutathione (GSH), S-adenosylmethionine (SAM) and S-adenosylhomocysteine (SAH) [31,56,57,58]. Decreases in GSH, SAM, and SAH levels cause global DNA hypomethylation [59,60], increase the risk of carcinogenesis [61,62], is associated with lipid peroxidation and cause age-related neurodegenerative diseases [63]. A decrease in SAH levels stimulates the DNMT1 enzyme [57] and contributes to the hypermethylation of CpG regions of tumor suppressor genes (for example *P53*) [31].

ROS also exerts harmful effects by activating secondary signaling pathways, for example, increases levels of interleukin 1β (IL1β), interleukin 6 (IL6), and tumor necrosis factor (TNF), and stimulates nuclear factor kappa B (NF-κB) [64,65], which indirectly increases the likelihood of malignant transformation [31,65,66,67], and is also directly proinflammatory [64,68,69]. TNF-α through IFN activation causes global DNA hypomethylation in aging cells [49,70]. Furthermore, when IL1β is present in high amounts, it stimulates additional inflammatory growth factors, namely TNF and matrix metalloproteinases (MMPs), etc. [71]. Both MMPs and TNF (in a redundant manner) promote malignant transformation of cells, as well as their progression [72], and activate NF-κB [71,73,74,75], thus forming a positive feedback loop. The mentioned interleukins and NF-κB mutually activate each other, and they also generate additional ROS [66,76], also forming a positive feedback loop.

Moreover, both DMBA and TFA activate the 3-hydroxy-3-methylglutaryl-coenzyme A reductase (HMG-CoAR) enzyme, which synthesizes cholesterol (for example in hepatocytes) that increases membrane rigidity [77,78]. For the sake of completeness, we need to mention that in the case of TFAs paradoxically, a decrease in cholesterol levels in Wistar rats has also been reported by Huang et al. [79]. With membrane rigidity and ROS formation, a positive association is presented within the phospholipid bilayer of the membrane [80] and ROS activity that elevates the risk of inflammation and malignant transformation [81,82]. For example, the increase of cholesterol levels in membranes favors the activation of the RAS oncogene family [78] both directly, through affecting the membrane rafts, and indirectly, via glycosylphosphatidylinositol (GPI) anchor proteins bound to membrane rafts [83,84].

The production of F2-isoprostane (F2-isoPs) increases up to 100-fold concentration in response to cholesterol and oxidative stress (predominantly lipid peroxidation) [81,85]. F2-isoPs distorts membrane fluidity and integrity [81]. Nevertheless, F2-isoPs increase the risk of carcinogenesis as well, for example by increasing proliferation [86]. Moreover, plasma free and total (free plus esterified) F2-isoPs increase with age (185% and 66%, respectively), but these increases are reduced by life-extending caloric restriction (50% and 23%, respectively) [87]. The levels of esterified F2-isoPs increase 68% with age in the liver, and 76% with age in the kidney, but caloric restriction modulated the age-related increase, reducing the esterified F2-isoPs levels 27% in the liver and 35% in the kidney [87].

### 4.2. Effect of DMBA on the L1-RTP DNA Methylation and Aging

DMBA caused significant L1-RTP DNA and oncogene (for example, *RAS* gene family) hypomethylation as well as hypermethylation of tumor suppressor genes (for example *P53*) compared to the control group via influencing DNMT enzymes [31,36,88]. Activated K-RAS hypermethylated the transcription factors of the tumor suppressor gene *INK4-ARF,* and thus silenced its expression [Struhl, 2014]. Its significance is that ARF/P53 signaling pathway is protective and has been shown to play an important role in slowing down aging [45], while P53 inhibits transposase enzyme [89] and hinders L1-RTP and presumably global DNA hypomethylation as well [31].

DMBA also activates the mitogen-activated protein kinase (MAPK) and Janus kinase (JAK) secondary signaling pathways [76], which activates the above-mentioned interleukins (and consequently NF-κB). These processes finally lead to global DNA hypomethylation [31] and accelerate aging, for example, by decreasing the expression of antitumorigenic microRNA-134 (miR-134) and *P53* [67,75,90,91,92].

DMBA also significantly elevated *mTORC1* gene expression and miR-9 level in the liver, spleen, and kidneys of CBA/CA female mice, compared to untreated controls [93]. DMBA activates the enzymes of glycolysis and lipogenesis [77]. Indeed, DMBA exposure in female Sprague-Dawley rats significantly elevated blood glucose levels compared to untreated controls [94]. The consequently released growth factors such as insulin and insulin-like growth factor (IGF) activate mTORC1 through phosphoinositide 3-kinase AKT-tuberous sclerosis-RHEB (PI3K-AKT-TSC-RHEB) signaling [95]. mTORC1 stimulates glycolysis and glucose uptake through modulating the transcription factor hypoxia-inducible factor (HIF1α) (Düvel 2010). HIF-1 increases glucose uptake and cell proliferation by increasing the expression of insulin-like growth factor 2 (IGF2) and *C-MYC* [96]. HIF-1 also induces inflammation by upregulating TNFα and cancer metastasis by upregulating fibronectin 1 [96]. However, increased activity of both mTOR and HIF-1 reduces life expectancy [96,97]. The expression level of miR-9 is increased by C-MYC, and miR-9 inhibits the progression of HCC as a tumor suppressor, but miR-9 also amplifies E-cadherin, which increases *C-MYC* expression, which increases miR-9 level, forming a positive feedback loop [93,98].

### 4.3. Effect of TFA on the L1-RTP DNA Methylation Pattern

TFA enters the cell membranes and increases their rigidity directly too leading to oxidative damage and inflammation [15]. Furthermore, TFAs decrease adiponectin and peroxisome proliferator-activated receptor gamma (PPAR-γ) activity [15,53]. If PPAR-γ is inactivated, it increases inflammatory response and hinders cholesterol transport, glucose, and fatty acid storage and promotes F2-isoPs formation [99]. Thus, the decrease of PPAR-γ activation results in a positive feedback loop with the mentioned harmful effects [53] [Smith, 2009], and it also hinders preadipocyte differentiation, thereby increasing the risk of developing malignant tumors and hinders tissue regeneration too [53,100].

Elaidic acid (trans-9-octadecenoic acid) (EA), induced global hypomethylation of THP-1 cells in vitro and activated proinflammatory (e.g., TNF-α, IL-6, C-reactive protein (CrP)) and adipogenic signaling pathways at concentrations of 50-200 μM [53,101,102]. Both trans-linoleic acid (trans, trans-9-12-octadecadienoic acid) (LA) and EA increase the levels of intercellular adhesion molecule-1 (ICAM-1) and vascular cell adhesion molecule-1 (VCAM-1) [64]. ICAM-1 and VCAM-1 also generate ROS, which activates NF-κB, which has a direct proinflammatory effect [64]. These oxidative and inflammatory damages are added to the effects of DMBA as mentioned earlier [55].

### 4.4. Protective Effect of OO

#### 4.4.1. The Effect of Fat-Soluble Substances of OO on the LINE-1 DNA Methylation Pattern

The cell membrane fluidity enhancing effects of MUFA and PUFA promote DNA methylation via the above-mentioned secondary signaling pathways (for example by decreasing NF-κB) [67,103]. Theoretically, the saturated fatty acids, due to their membrane rigidity enhancing effects [104], could cause hypomethylation of L1-RTP DNA. In contrast, palmitic acid caused global hypermethylation [105] and reduced inflammation through the induction of the *PPARγ* gene [106] in human myocytes.

Oleic acid decreased the expression of *TNF-α* and *IL1β* and increased the anti-inflammatory *IL10* in septic mice [107]. Furthermore, oleic acid can also stimulate PPAR [108], which activates antioxidant response, has anti-inflammatory and neuroprotective effects [107,109].

Oleic acid between 1 mM and 150 mM concentration allosterically activates the NAD-dependent deacetylase sirtuin-1 (SIRT1) [110], which is a regulator of mTOR [Ghosh, 2010]. SIRT1 inhibits the DNMT1 enzyme and through inhibition of DNMT3L protein, it blocks the gene expression of *DNMT3A* and *DNMT3B* enzymes too, [111]. For example, in MDA-MB-231 breast cancer cell line, SIRT1 reduced the inhibitory effect exerted by DNMT1 on tumor suppressor genes *ERα* and *CDH1* [112]. (Interestingly, *DNMT3* blocking effect was not accompanied by a decrease in the activity of the enzymes [111], but synergically with other chemopreventive agents, this could be still relevant).

PUFA, through its direct β-catenin inhibitory effect significantly reduced the expression of DMBA-induced *C-MYC* oncogene, compared to controls [44,52,113]. This is relevant, with respect to the DNA methylation pattern, is high since C-MYC induces oncogenic expression of the ten-eleven translocation methylcytosine dioxygenase 1 (TET1) gene, which codes for a DNA demethylating protein [114].

In addition, SIRT1 inhibits oxidative-stress-associated cellular aging [97], and C-MYC as well, through inhibition of β-catenin, which is important in the liver [67,115]. (In contrast, SIRT1 also inhibits P53, and hence SIRT1 may also act as an oncogene [115]). Moreover, oleic acid also prevented TNF-induced decline in insulin level by promoting the translocation of the transcription factor PPARγ into the nucleus, in a male KKAy type II diabetic mouse model [116].

#### 4.4.2. Water-Soluble Substances of Olive Oil

Oleuropein and oleocanthal are water-soluble polyphenols of OO and are absorbed from the small intestine and reach the spleen and liver [117], where they exert a protective effect against ROS [118,119], mainly on the cell membrane [120].

Oleuropein can inhibit the activation of NF-κB [56,121] and increase the intracellular level of GSH, which is protective against the harmful effects of ROS [122,123,124]. Furthermore, oleuropein is also a PPARα agonist anti-inflammatory constituent [106,125]. 

Oleocanthal is a potent inhibitor of mTOR [126]. EVOO consumption significantly reduced the expression of mTORC1 gene both in the liver and the spleen of DMBA-treated CBA/Ca female mice [67]. Nanda et al. in Sprague-Dawley rats induced the DNMT1 enzyme by dimethylhydrazine and hypomethylated the promoters of *NFκB*, *MMP-9,* and *VEGF*, significantly increasing their gene expression compared to untreated controls, but these effects were counteracted by EVOO consumption [Nanda, 2019]. Indeed, the decrease in *DNMT1* expression demethylates the promoter region of phosphatase and tensin homolog (PTEN), leading to the decrease of mTOR expression [127]. Although SAM, derived from methyl donor, stimulates mTOR through the SAMTOR protein [34], this effect is ultimately counteracted by EVOO [67].

### 4.5. L1-RTP DNA Methylation Patterns

ROS induce elevated blood glucose level, which is reflected in age-dependent biomarkers of renal damage, such as oxidant-sensitive heme oxygenase, advanced glycation end product (AGE), and F2-isoPs [128]. However, F2-isoPs, when added in vitro to renal mesangial cells (under high glucose levels, to which DMBA also contributes [94]; see above), increased the gene expression of TGF-β by activating protein kinase-C (PKC) [129]. TGF-β induced both expression and activity of DNA methyltransferases (DNMT) -1, -3A, and -3B in ovarian cancer cells [130], while in vitro phosphorylation of DNMT1 by PKCζ reduced its methyltransferase activity [131]. TGFβ, as a tumor suppressor, acts as a double-edged sword and activates anti-inflammatory signaling, but when its receptor loses function during malignant transformation, it indirectly acts as an immunosuppressant, promoting vascularization and metastasis, and thus enhances the malignancy of carcinomas [132] as mentioned earlier [20]. 

EVOO significantly decreased the DMBA-induced L1-RTP DNA hypomethylation both in the liver and spleen but not in the kidneys of experimental animals. This may be related to the fact that hypomethylation of L1-RTP DNA is not common even in RCC [133]. TFA tends to incorporate into the kidneys in smaller amounts than into the liver [79]. In Wistar rats, Huang et al. measured 1.2 mg/g TFA in the liver and only 0.6 mg/g TFA in the kidneys after their 16 weeks of consumption of a diet containing 4.5% TFA [79]. Indeed, lipid sensitivity of organs and hypomethylation of the L1-RTP DNA segment are associated in the case of TFA exposure [134].

#### 4.5.1. L1-RTP DNA Methylation Pattern in the Liver and Spleen

The trans-3-hexadecenoic acid significantly increased the *mTOR* gene expression in the liver of DMBA treated mice group, even compared to the increase induced by DMBA exposure [67,77]. This can be explained by the fact that TFA inhibits the activity of CAT, SOD, and GSH peroxidase enzymes in lipid-sensitive liver and spleen [78,134]. Furthermore, TFA depletes antioxidant molecules (for example, GSH), which mainly protects against hepatotoxic processes [58]. Thereby TFA indirectly promotes the above-mentioned inflammation, tumor formation, and global DNA hypomethylation [31]. In addition, the elevated F2-isoPs levels under DMBA and TFA damage enhance the proliferation of stellate cells in the liver [86].

In nonalcoholic steatohepatitis (NASH) diseases, which include liver fibrosis and liver cancer, the composition of the cell membrane and the PPARα and the methylation pattern of DNA is also important [105,135]. Oleuropein as a PPARα agonist exerts hepatoprotective effects, such as reducing triglyceride levels [125]. Indeed, in hepatocellular carcinomas (HCC) the adenomatous polyposis coli (*APC)* and *RASSF1* tumor suppressor genes were hypermethylated and the *MEST* gene was hypomethylated [136]. Both APC and RASSF1 slows cell proliferation—the former inhibits β-catenin, while the latter induces a cell cycle arrest mechanism by inhibiting cyclin D1, while MEST phosphorylates and thereby activates the transcription factor CREB, which enhances the expression of the *C-FOS* proto-oncogene [137]. Its importance is that in healthy aging, exons 1 and 4 of the *C-FOS* gene are hypermethylated, but both liver cirrhosis and liver carcinogenesis are accompanied by hypomethylation [138]. In an in vivo rat model, the DMBA and corn oil induced hypermethylation of *RASSF1* promoter, but it was ameliorated by EVOO through decreasing DNMT1 enzyme’s activity [139]. Even in ApcMin/+ mice (that spontaneously develop intestinal polyps), the OO-enriched diet reduced polyp number and volume through a reduction of proliferation as well as proapoptotic effect by inhibiting fatty acid synthase and HMGCoA reductase gene expression [140]. Intriguingly, the secoiridoid polyphenol content of EVOO activated through C-FOS pathway the AP-1 (activator protein-1) transcription factors, which in this context were not associated with tumorigenesis but rather with growth inhibition and/or differentiation of breast cancer cells [141]. The predominant antiaging effect of EVOO secoiridoids was exerted through inhibiting mTOR and not by decreasing C-FOS activity [141].

PPARγ also regulates inflammatory factors in the liver, but the promoter of *PPARγ* is hypermethylated both in liver inflammation and liver fibrosis, and thus its expression is reduced [142]—although the oleuropein content of EVOO can counteract it [106]. Moreover, in rats fed with a high-fat diet, EVOO prevented hyperglycemia, insulinemia, apoptosis of pancreatic β-cells, and improved insulin resistance [143].

#### 4.5.2. L1-RTP DNA Methylation Pattern in the Kidneys

The result of the kidneys, namely that the DMBA (or DMBA+TFA) induced L1-RTP DNA hypomethylation was weaker than in the other examined organs, could be explained by decreased DMBA damage through the generally silenced *TSPYL5* gene [144] and by the generally increased expression of the antioxidant and anti-inflammatory *lactoferrin* (*LTF*) gene in the kidneys [144,145].

Both the expression of the *TSPYL5* gene and the amount of TSPYL5 protein decrease with age [144] because both the DNMT1 (also indirectly activated by DMBA [31]) and the DNMT3B enzymes can cause hypermethylation of the promoter region of the *TSPYL5* gene [146]. The TSPYL5 inhibits the activity of ubiquitin-specific protease 7 (USP7), which is the deubiquitylase enzyme for the P53 [147]. In summary, TSPYL5 reduces the activity of USP7 toward P53, resulting in increased P53 degradation through ubiquitylation [147]. Thus, ultimately, the decrease of *TSPYL5*, which inhibits the P53 and P21 tumor suppressors, may be the cause of the reduction of risk of mutation in the kidneys, compared to other organs. Indeed, *P53* expression is slightly increased due to DMBA treatment in CBA/Ca mice in comparison to corn oil control [52] and P53 promotes both global and L1-RTP DNA hypermethylation by inhibiting LINE-1 transposons [89,148,149]. For the sake of completeness, it should be mentioned that TSPYL5 gene hypermethylation also occurs in HCC cells [150] as a protective mechanism.

*LTF* is generally highly expressed in the human kidneys, increasing further with age and is in vivo protective against DMBA generated ROS damage [144,145,151]. However, LTF from the viewpoint of senescence, as a double-edged sword, not only suppresses ROS-induced senescence of human mesenchymal stem cells (hMSCs) but also activates NF-κB through the Toll-like receptor 4 pathway [56,152].

The difference between the result in the kidneys and in other studied organs are explained by the fact that OO does not induce oxidative stress in the kidneys, but does in the other examined organs [153]. Thus, in the kidneys, the expression of the stress-dependent *P53* gene was only slightly increased due to DMBA treatment [Budan, 2009; Kouka, 2020], while P53 could have theoretically stimulate global and L1-RTP DNA methylation, as mentioned earlier [89,148,149].

The effect of EVOO on methylation pattern may also contribute to the decrease in *TSPYL5* expression and to the increase in *LTF* gene expression, which explains the reduced possibility of somatic mutation proportional to aging observed in the KIRC and the KIRP cell lines [Horvath, 2013], which is supported by the methylation pattern of the renal L1-RTP DNA in this study (Figure 3).

## 5. Conclusions

Both DMBA treatment and DMBA added combined with TFA caused significant L1-RTP DNA hypomethylation in the liver, spleen, and kidneys of CBA/Ca mice. According to the literature, DMBA forms DNA adducts and thereby inhibits tumor suppressor genes (for example, *P53*), activates oncogenes (for example, *RAS*, *C-MYC, BCL-2, NOTCH*), and alters microRNA (for example miR-9, miR-124, miR-132; miR-134) patterns leading to global hypomethylation [29,31,52,67,93,154].

Both DMBA and TFA manifest a dominant oxidative stress source by generating ROS and exerts proinflammatory effect, with mostly overlapping molecular biological features, namely depleting antioxidants (for example, GSH, SAM, SAH) promoting inflammatory signaling pathways (for example, IL-1β, IL-6, TNF, NF-κB, mTOR), and causing ultimately L1-RTP DNA hypomethylation [31,67].

Especially important is that according to the literature, TFA decreases PPAR-γ activity [Ali Abd El-Aal, 2019; Smith, 2009], which could otherwise ameliorate the harmful effect of DMBA [155], but if one is exposed to both agents, the synergically deleterious effect of DMBA and TFA exacerbates L1-RTP DNA hypomethylation, as reflected in the results of the present study. Moreover, TFA administration combined with DMBA further increased the significant L1-RTP DNA hypomethylation due to increased oxidative stress as well as increased adipogenic secondary signal transducers induction. Since aging and L1-RTP DNA methylation are similar in human and mouse species [156], the results are also of human relevance [157].

EVOO exerts antioxidant and anti-inflammatory effects directly on cell membranes, and through the regulation of secondary signal transporters [11,12,67], DMBA decreased significantly; additionally, combined DMBA + TFA-induced L1-RTP DNA hypomethylation was observed in the liver and spleen but not significantly in the kidneys of CBA/Ca mice. EVOO induces the *PPARγ* gene [106], and thereby, theoretically, it could decrease the mentioned synergic damage of DMBA combined with TFA.

In summary, high EVOO intake with diet decreases the likelihood of cancer and increases life expectancy because EVOO can counteract DMBA and TFA-induced damage by improving global DNA methylation pattern, while decreasing hyperglycemia, mTOR activity, and inducing SIRT1 function among other [11,12,97,103,106,158] (Figure 4).

## Figures and Tables

**Figure 1 nutrients-14-00908-f001:**
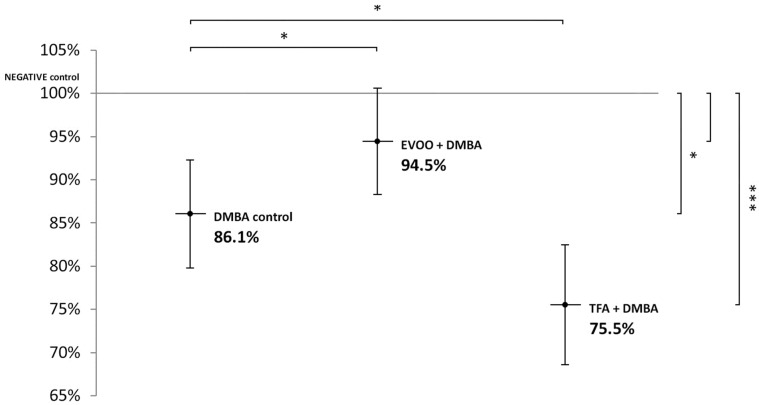
L1-RTP DNA methylation pattern in the spleen of CBA/Ca female mice (*n* = 6) exposed to the effect of DMBA, and the effect of EVOO or TFA coadministered with DMBA, expressed as the percentage of untreated control (* *p* < 0.05; *** *p* < 0.001). L1-RTP DNA: LINE-1 retrotransposon deoxyribonucleic acid, EVOO: extra virgin olive oil, TFA: trans-fatty acid.

**Figure 2 nutrients-14-00908-f002:**
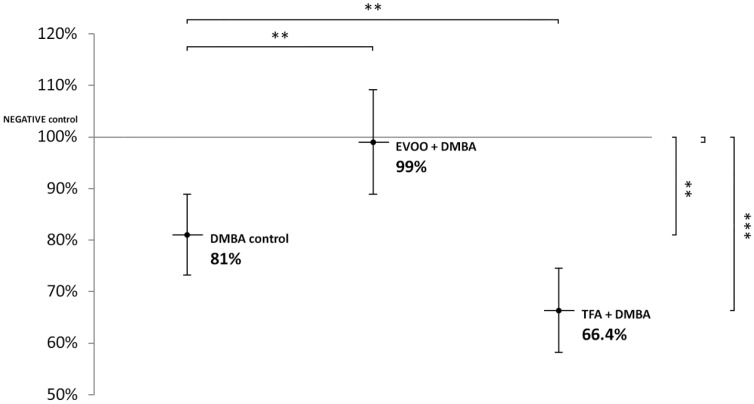
L1-RTP DNA methylation pattern in the liver of CBA/Ca female mice (*n* = 6) exposed to the effect of DMBA, and the effect of EVOO or TFA coadministered with DMBA, expressed as the percentage of untreated control (** *p* < 0.01; *** *p* < 0.001).

**Figure 3 nutrients-14-00908-f003:**
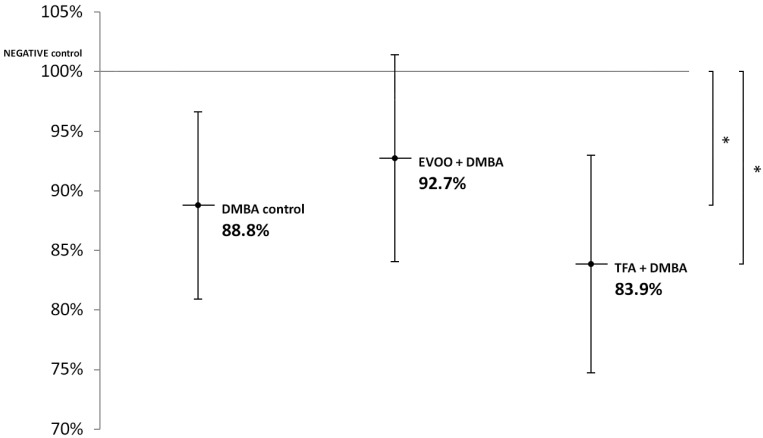
L1-RTP DNA methylation pattern in the kidneys of CBA/Ca female mice (*n* = 6) exposed to the effects of DMBA and the effects of DMBA + EVOO or DMBA + TFA, expressed as the ratio of untreated control (* *p* < 0.05).

**Figure 4 nutrients-14-00908-f004:**
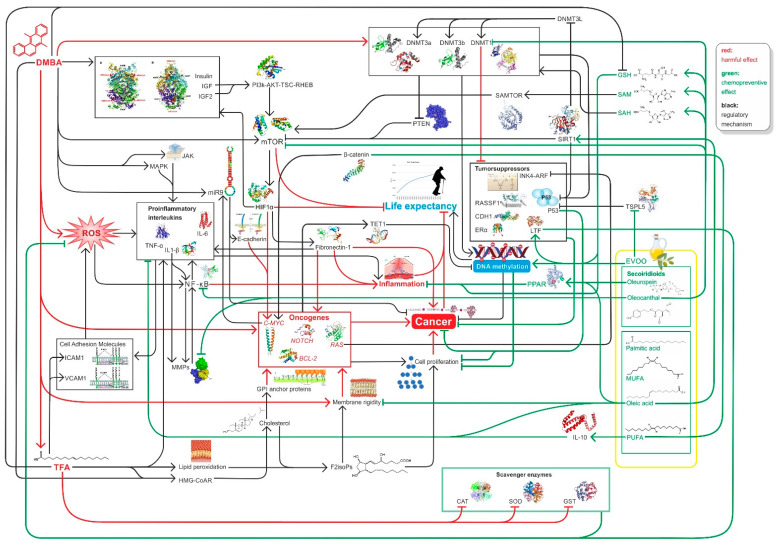
Summary of relevant factors influencing inflammation, carcinogenesis DNA methylation, and ageing in connection with our study.

**Table 1 nutrients-14-00908-t001:** Treatment and feeding of the study groups.

Name of the Group	ip. DMBA	Daily Dose/Animal	Manufacturer	Latin/Scientific Names
negative control	–			
positive control	+		Sigma Aldrich Ltd.	dimethylbenz[a]anthracene
EVOO	+	0.3 g	Agraria Riva Del Garda SCA	*Oleum virgineum*
TFA	+	0.3 g	Sigma Aldrich Ltd.	trans-3-hexadecenoic acid

## Data Availability

Data are available upon reasonable request from the corresponding authors.

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
