# Peer review of "Olive Oil Improves While Trans Fatty Acids Further Aggravate the Hypomethylation of LINE-1 Retrotransposon DNA in an Environmental Carcinogen Model"

_nutrients, 2022, doi:10.3390/nu14040908_

Round 1

Reviewer 1 Report

In the manuscript authors used extra virgin olive oil and trans fatty acids to supplement mice' diet. The effect of 7,12-dimethylbenz[a]anthracene on DNA methylation in the spleen, liver and kidneys was studied. Results are broadly discussed.

Author Contribution section is needed.

Author Response

Dear Reviewer 1,

authors thank your kind notifications.

QUESTION AND COMMENT OF REVIWER:

„In the manuscript authors used extra virgin olive oil and trans fatty acids to supplement mice' diet. The effect of 7,12-dimethylbenz[a]anthracene on DNA methylation in the spleen, liver and kidneys was studied. Results are broadly discussed.

Author Contribution section is needed.”

AUTHOR’S ANSWER:

            The text is compared with author contributions:

„Author Contributions

Conceptualization, László Szabó and Ferenc Budán; Data curation, Andras Tomesz, Richard Darago and Timea Varjas; Formal analysis, Andras Tomesz, Arpad Deutsch and Timea Varjas; Funding acquisition, Istvan Kiss; Investigation, László Szabó, Richard Molnar, Andras Tomesz, Richard Darago and Timea Varjas; Methodology, László Szabó, Richard Molnar, Arpad Deutsch, Richard Darago, Zsombor Ritter, Jozsef L. Szentpeteri, Domokos Máthé and Ferenc Budan; Project administration, László Szabó, Richard Molnar, Domokos Máthé, Attila Sik, Ferenc Budan and Istvan Kiss; Resources, László Szabó, Zsombor Ritter, Jozsef L. Szentpeteri, Kitti Andreidesz, Attila Sik and Ferenc Budan; Software, Andras Tomesz, Richard Darago and Zsombor Ritter; Supervision, Zsombor Ritter, Imre Hegedüs, Attila Sik and Istvan Kiss; Validation, Arpad Deutsch, Kitti Andreidesz, Domokos Máthé and Attila Sik; Visualization, Jozsef L. Szentpeteri, Kitti Andreidesz, Domokos Máthé and Imre Hegedüs; Writing – original draft, László Szabó, Ferenc Budán, Domokos Máthé; Writing – review & editing, Imre Hegedüs and Istvan Kiss.”

Reviewer 2 Report

The manuscript is very interesting. There are many points to review, in order to make it more uniform. I suggest the following changes.
1- Introduction: This point is very long. I suggest providing the essential contents.
2- Discussion:  This point is a bit distracting. I suggest to review.
3- References: the references should be fixed. They begin with the number ‘48’.

Author Response

Dear Reviewer 2,

authors thank your kind notifications.

QUESTION AND COMMENT OF REVIWER:

„1- Introduction: This point is very long. I suggest providing the essential contents.”

AUTHOR’S ANSWER:

The Introduction was reviewed and some sentences were changed. For details please find the attached revised manuscript file.

QUESTION AND COMMENT OF REVIEWER:

„2- Discussion:  This point is a bit distracting. I suggest to review.”

AUTHOR’S ANSWER:

            The full Discussion part was reviewed and some sentences were removed or changed.

Please, find the attached revised manuscript file.

QUESTION AND COMMENT OF REVIEWER     

“3- References: the references should be fixed. They begin with the number ‘48’.”

AUTHOR’S ANSWER:

            The full reference list was reviewed and numerous references are placed on it.

            Please, find the attached revised manuscript file.
